# Role of Balanced Involvement of the ICOS/ICOSL/Osteopontin Network in Cutaneous Wound Healing

**DOI:** 10.3390/ijms252212390

**Published:** 2024-11-19

**Authors:** Foteini Christaki, Amirabbas Ghasemi, Deepika Pantham, Reza Abouali, Alessia Provera, Cristina Vecchio, Anteneh Nigussie Sheferaw, Chiara Dianzani, Salvatore Sutti, Roberta Rolla, Sara Sacchetti, Luca Giacomini, Umberto Dianzani, Ian Stoppa

**Affiliations:** 1Department of Health Sciences, Università del Piemonte Orientale, 28100 Novara, Italy; foteini.christaki@uniupo.it (F.C.); 20052072@studenti.uniupo.it (A.G.); deepika.pantham@uniupo.it (D.P.); 20046595@studenti.uniupo.it (R.A.); alessia.provera@uniupo.it (A.P.); cristina.vecchio@uniupo.it (C.V.); 20046466@studenti.uniupo.it (A.N.S.); salvatore.sutti@med.uniupo.it (S.S.); roberta.rolla@med.uniupo.it (R.R.); sara.sacchetti@uniupo.it (S.S.); 20023552@studenti.uniupo.it (L.G.); ian.stoppa@uniupo.it (I.S.); 2Department of Scienza e Tecnologia del Farmaco, University of Turin, 10124 Turin, Italy; chiara.dianzani@unito.it

**Keywords:** ICOS/ICOSL/OPN network, wound healing, reparative macrophages

## Abstract

Inducible T-cell costimulator (ICOS, CD278) is a costimulatory receptor primarily expressed by activated T cells. It binds to ICOS ligand (ICOSL, CD275), which is expressed by various immune and non-immune cell types, particularly in inflamed tissues. ICOSL can also bind to osteopontin (OPN), a protein that functions both as a component of the extracellular matrix and as a soluble pro-inflammatory cytokine. Previous studies, including ours, have shown that ICOS and ICOSL play a role in skin wound healing, as mice deficient in either ICOS or ICOSL exhibit delayed healing. The aim of this study was to investigate the involvement of the ICOS/ICOSL/OPN network in skin wound healing by analyzing mice that are single knockouts for ICOS, ICOSL, or OPN, or double knockouts for ICOS/OPN or ICOSL/OPN. Our results showed that wound healing is impaired in all single knockout strains, but not in the two double knockout strains. Cellular and molecular analyses of the wound healing sites revealed that the healing defect in the single knockout strains is associated with reduced neutrophil infiltration and decreased expression of α-SMA (a marker of myofibroblasts), IL-6, TNFα, and VEGF. In contrast, the normalization of wound closure observed in the double knockout strains was primarily linked to increased vessel formation. A local treatment with recombinant ICOS-Fc improved healing in all mouse strains expressing ICOSL, but not in those lacking ICOSL, and led to a local increase in vessel formation and macrophage recruitment, predominantly of the M2 type.

## 1. Introduction

Skin wound healing begins immediately after injury and progresses through three distinct phases. The first is the inflammatory phase, characterized by platelets aggregation and the recruitment of inflammatory cells to the wound site. The second phase, known as the proliferative phase, is characterized by the formation of granulation tissue and re-epithelialization, driven by the migration and proliferation of keratinocytes, fibroblasts, and endothelial cells, as well as with extracellular matrix deposition. The final phase, the remodeling phase, completes the regenerative process with the reorganization of connective tissue to promote scar formation [1,2].

Inducible T-cell costimulator (ICOS, CD278) is a costimulatory receptor primarily expressed by activated T cells [3,4]. It binds to the ICOS ligand (ICOSL, CD275), which is expressed by various immune and non-immune cell types, especially in inflamed tissues [5,6]. On the one hand, the ICOS engagement by ICOSL enhances cytokine production in T helper cells, boosts cytotoxic activity in cytotoxic T cells and NK cells, and contributes to the differentiation of regulatory T (Treg) cells [3,7]. On the other hand, the ICOSL engagement by ICOS modulates the activity of ICOSL-expressing cells, influencing cell migration, cytokine secretion, antigen presentation, and angiogenesis [8,9,10,11,12]. Furthermore, ICOSL can bind to osteopontin (OPN), a protein that functions both as a component of the extracellular matrix and as a soluble pro-inflammatory cytokine [13]. ICOS and OPN bind to the non-overlapping sites on ICOSL and exert distinct, often opposing effects on the ICOSL-expressing cells [14]. For example, the OPN-mediated activation of ICOSL promotes the migration of various tumor cell types and enhances tumor angiogenesis, while the ICOS-mediated activation inhibits these processes [14,15].

It has been reported by our group and others that ICOS and ICOSL are involved in skin wound healing since mice deficient in either ICOS (ICOS-KO) or ICOSL (ICOSL-KO) or both (ICOS/ICOSL-KO) display delayed healing [16,17]. Maeda et al. showed that, in these knockout mice, this delay is accompanied by several defects in the healing site, including decreased infiltration of T cells and macrophages, delayed angiogenesis, and decreased production of IL-4, IL-10, and IL-6. These authors suggested that the defective production of IL-6 may play a key role in the healing delay and that it might be ascribable to the defective ICOS-mediated costimulation of the type 2 T helper (Th2) cells. Tissue repair was indeed improved either by a local treatment with IL-6 or, in ICOS-deficient but not in ICOSL-deficient mice, by an adoptive transfer of wild type T cells expressing ICOS [16]. Therefore, these data highlighted the attention on the defective activity of ICOS on T cells, which is shared by both ICOS-KO and ICOSL-KO mice. Subsequently, we showed that also the defective activity of ICOSL on the ICOSL-expressing cells may play a role in the delayed healing. This was evidenced by the fact that the local treatment of the wounds with ICOS-Fc, a soluble recombinant protein formed by the fusion of the ICOS extracellular portion with the IgG1 Fc portion and capable of triggering ICOSL independently from the endogenous ICOS, improves wound healing in wild type and ICOS-KO mice, but not in ICOSL-KO mice [17]. In wild type mice, the treatment with ICOS-Fc induces substantial changes in the healing tissue, including an increase in T cells, M2 macrophages, vessels, and fibroblasts, a decrease in neutrophils, and increased expression of IL-6 and VEGF [17]. These data are in line with data obtained in the acute liver damage induced by CCl_4_, showing that the ICOS-KO mice display a severe impairment of recovery from liver damage, accompanied by a local decrease in TREM2^+^ reparative monocyte-derived macrophages (MoMFs) and ICOS^+^CD8^+^ T cells; these defects are all restored by the treatment with ICOS-Fc [18]. The aim of this work was to further analyze the involvement of the ICOS/ICOSL/OPN network in skin wound healing by comparing healing in mice that are single knockout for either ICOS (ICOS-KO) or ICOSL (ICOSL-KO) or OPN (OPN-KO) and mice double knockout for either ICOS and OPN (ICOS/OPN-KO) or ICOSL and OPN (ICOSL/OPN-KO). The results show that wound healing is impaired in all single-KO mouse strains, but not in the double knockout strains. Cellular and molecular analyses of the healing tissue show that the healing defect of the single-KO strains is accompanied by the defective infiltration of neutrophils and defective expression of α-SMA, IL-6, TNFα, and VEGF. In the double-KO strains, normalization of the wound closure is mainly accompanied by an increase in vessel formation. A local treatment with ICOS-Fc improves healing in all strains expressing ICOSL, accompanied by an increase in vessels and macrophages, possibly of the M2 type, whereas no effect is detected in the mouse strains not expressing ICOSL.

## 2. Results

### 2.1. Wound Healing in the Different Mouse Strains

#### 2.1.1. Wound Closure

To investigate the role of the ICOS/ICOSL/OPN network in wound healing, skin wound healing was compared in wild type (WT), ICOS-KO, ICOSL-KO, OPN-KO, ICOS/OPN-KO, and ICOSL/OPN-KO mice. The results showed that, compared to WT mice, wound healing was delayed in ICOS-KO, ICOSL-KO, and OPN-KO mice, whereas the healing kinetics were surprisingly normal in ICOS/OPN-KO and ICOSL/OPN-KO mice (Figure 1).

#### 2.1.2. Histological Analyses

Histological analyses of the healing sites were performed at day 3 (T3) and 4 (T4) after the wound by evaluating the fibroblast levels by hematoxylin and eosin (H&E) and collagen fiber deposition by picrosirius red staining. Moreover, blood vessels formation and the recruitment of T cells, neutrophils, and macrophages were evaluated by the immunohistochemical staining of CD31, CD3, MPO, and F4/80, respectively. The results were analyzed by comparing the differences, at either T3 or T4, between WT and each KO strain (Figure 2). In detail, Figure 2A shows that the fibroblast levels were not significantly altered in all single-KO strains, but they were decreased in both double-KO strains (at T4). Figure 2B indicates that the collagen levels were not significantly changed in the single KO strains, but they were decreased in ICOSL/OPN-KO mice (at T3 and T4), and increased in ICOS/OPN-KO mice (at T4). Figure 2C displays data regarding the vessel formation that was not significantly altered in OPN-KO and ICOS-KO mice, but it was increased in ICOSL-KO mice (at T4) as well as in ICOSL/OPN-KO and ICOS/OPN-KO mice (both at T3 and T4). Figure 2D shows that the presence of macrophages was not significantly changed in ICOS-KO, OPN-KO, and ICOS/OPN-KO mice, while it was decreased in ICOSL-KO mice (at T3 and T4), and increased in ICOSL/OPN-KO mice (at T3). Figure 2E shows that the neutrophil levels were not significantly changed in ICOSL/OPN-KO mice, but they were decreased in all the other KO strains (at T3 and T4). Figure 2F shows that the T cells were not significantly changed in any knockout strain except for an increase in OPN-KO mice (at T4).

#### 2.1.3. Molecular Analyses

To better characterize the inflammatory and fibrogenic microenvironment of the wound healing sites, we assessed the mRNA levels of IL-6, VEGF, TNFα, α-SMA (marking myofibroblasts), TREM1 and TREM2 (marking M1 and M2 macrophages, respectively) at day 1, 2, and 3 by real-time PCR. The results were analyzed by comparing the differences, at each time point, between WT and each KO strain (Figure 3). Figure 3A shows that α-SMA was significantly decreased in all KO strains; at T3 in ICOS-KO, ICOSL-KO, and ICOSL/OPN-KO mice, and at all time points in OPN-KO and ICOS/OPN-KO mice. Figure 3B shows that IL-6 was decreased in all KO strains; at T3 in ICOS-KO, ICOSL-KO, and OPN-KO mice, at T1 and T3 in ICOSL/OPN-KO mice, and at all time points in ICOS/OPN-KO mice. Figure 3C shows that VEGF was not significantly changed in ICOS/OPN-KO mice, whereas it was decreased in ICOSL/OPN-KO (at T2 and T3), and ICOS-KO, ICOSL-KO, and OPN-KO mice (at T2). Figure 3D shows that TNFα was significantly lowered in OPN-KO and ICOSL/OPN-KO mice (at all time points), and ICOS-KO and ICOSL-KO mice (at T1 and T3). In contrast, it was decreased at T1 and T3 and increased at T2 in ICOS/OPN-KO mice. Finally, the TREM2/TREM1 ratio was not significantly changed in any knockout strain.

### 2.2. Effects of ICOS-Fc in the Different Mouse Strains

#### 2.2.1. Wound Closure

In a previous work [17], we showed that a local instillation of ICOS-Fc in the wound improves wound closure in WT and ICOS-KO mice, but not in ICOSL-KO and ICOS/ICOSL-KO mice, indicating that the ICOS-Fc effect requires the expression of ICOSL. To discriminate the role of the different components of the ICOS/ICOSL/OPN network in the ICOS-Fc effect, we extended the analysis to OPN-KO, ICOS/OPN-KO, and ICOSL/OPN-KO mice. The results showed that ICOS-Fc favored wound closure also in OPN-KO and OPN/ICOS-KO mice, but not in ICOSL/OPN-KO mice (Figure 4), confirming that the ICOS-Fc effect is dependent on the expression of ICOSL.

#### 2.2.2. Histological Analyses

Our past work [17] illustrated that, in WT mice, the treatment of skin wounds with ICOS-Fc increases blood vessels and fibroblasts, and the infiltration of T cells and macrophages, whereas it decreases neutrophils [17]. To discriminate the role of the different components of the ICOS/ICOSL/OPN network in the ICOS-Fc effects, we extended the histological analyses to OPN-KO, ICOS/OPN-KO, and ICOSL/OPN-KO mice treated or not with ICOS-Fc. The results were analyzed by comparing the differences, at each time point, between the wounds treated with PBS (control) or ICOS-Fc. As expected, ICOS-Fc did not exert significant effects on any of these parameters in the mouse strains deficient in ICOSL, i.e., ICOSL-KO and OPN/ICOSL-KO mice. In contrast, it exerted variable effects in the mouse strains expressing ICOSL, i.e., ICOS-KO, OPN-KO, and OPN/ICOS-KO (Figure 5). Figure 5A shows that the treatment with ICOS-Fc significantly increased the fibroblast levels in ICOS-KO and OPN-KO mice at T3 and T4, similarly to WT mice. In contrast, it decreased the fibroblasts in OPN/ICOS-KO mice at T3. Figure 5B,C show that the treatment with ICOS-Fc significantly increased the number of vessels and macrophages in ICOS-KO, OPN-KO, and ICOS/OPN-KO mice, similarly to WT mice [17]. Specifically, the blood vessel numbers were increased at T3 and T4 in ICOS-KO and ICOS/OPN-KO mice, and at T3 in OPN-KO mice; the macrophages were increased at T3 and T4 in ICOS-KO and OPN-KO mice, and at T3 in ICOS/OPN-KO mice. Figure 5D shows that the treatment with ICOS-Fc decreased the neutrophil levels in ICOS/OPN-KO mice at T3 and T4, similarly to WT mice. In contrast, it increased the neutrophil levels in ICOS-KO and OPN-KO mice at T3 and T4. Figure 5E shows that the treatment with ICOS-Fc increased the number of T cells in ICOS-KO and ICOS/OPN-KO mice at T3 and T4, similarly to WT mice. In contrast, it decreased the T cell levels in OPN-KO mice at T4.

#### 2.2.3. Molecular Analyses

The previous work showed that, in WT mice, the treatment of skin wounds with ICOS-Fc increased the local expression of α-SMA, IL-6, VEGF, and TREM2/TREM1 ratio, and decreased the TNFα levels [17]. Therefore, we analyzed the effect of the treatment with ICOS-Fc in the KO mouse strains. The results were analyzed by comparing the differences, at each time point, between the wounds treated with PBS (control) or ICOS-Fc. The results showed that the treatment with ICOS-Fc did not exert any effect in ICOSL-KO and ICOSL/OPN-KO mice, as expected. In contrast, it exerted variable effects in ICOS-KO, OPN-KO, and OPN/ICOS-KO mice (Figure 6). Figure 6A shows that the treatment with ICOS-Fc significantly increased the α-SMA expression in ICOS-KO and OPN-KO mice at T3, similarly to WT mice. In contrast, it had no effect on OPN/ICOS-KO mice. Figure 6B shows that the treatment with ICOS-Fc significantly increased the IL-6 expression in OPN-KO mice at T3 and ICOS/OPN-KO mice at T2, similarly to WT mice. In contrast, it decreased IL-6 in ICOS-KO mice at T1 and T3. Figure 6C shows that the treatment with ICOS-Fc significantly increased the VEGF expression in ICOS-KO mice at T1 and T3, similarly to WT mice. In contrast, it decreased VEGF in OPN/ICOS-KO mice at T2 and had no significant effect in OPN-KO mice. Figure 6D shows that the treatment with ICOS-Fc decreased the TNFα expression in OPN/ICOS-KO mice at T1 and T2, similarly to WT mice. In contrast, it increased TNFα in OPN-KO mice at T1 and had no significant effect in ICOS-KO mice. Figure 6E shows that the treatment with ICOS-Fc increased the TREM2/TREM1 expression ratio in ICOS-KO mice at T2, OPN-KO mice at T2 and T3, and ICOS/OPN-KO mice at T1, T2, and T3.

## 3. Discussion

This study demonstrates that all components of the ICOS/ICOSL/OPN network are involved in wound healing since a deficiency in each of them impairs the healing process as shown in ICOS-KO, ICOSL-KO, and OPN-KO mice. In contrast and surprisingly, the healing closure was apparently normal in ICOS/OPN-KO and ICOSL/OPN-KO mice, suggesting that the combination of the OPN deficiency with the deficiency in either ICOS or ICOSL has compensatory effects on wound healing.

The signaling pathways serving the ICOS/ICOSL/OPN network involve those activated by the ICOS- or OPN-mediated triggering of ICOSL; those activated by the ICOSL-mediated triggering of ICOS; and those activated by the OPN-mediated triggering of the various cell receptors bound by OPN, including several integrins and CD44 (Figure 7). Each of these pathways exerts several effects in multiple cell types since ICOSL and the other OPN receptors are widely expressed in immune and non-immune cells. ICOS itself, for a long time considered to be selectively expressed in T cells, has been found also in other cell types such as dendritic cells, macrophages, and plasma cells [19,20]. The global network of these pathways is differently unbalanced in each single-KO strain (Figure 7D–F), and it would be mostly silenced in the double-KO mice (Figure 7B,C). Therefore, these data suggest that wound healing is impaired when the functioning of the ICOS/ICOSL/OPN network is unbalanced but not when it is globally defective.

A previous study conducted by Maeda et al. showed that the wound healing defect of ICOS-KO and ICOSL-KO mice is accompanied by a decrease in vessel formation, presence of α-SMA^+^ myofibroblasts, and reduced infiltration of T cells, macrophages and, in the first few hours, neutrophils [16]. Moreover, these authors detected the reduced expression of IL-6 and suggested that it might be ascribable to the defective costimulation of Th2 cells through the ICOS/ICOSL dyad. In a subsequent work, we showed that the local application of ICOS-Fc improved wound healing in WT mice and this effect was accompanied by an increase in vessels, fibroblasts, T cells, and macrophages, a decrease in neutrophils, and increased expression of α-SMA, IL-6, VEGF, and the TREM2/TREM1 ratio.

Monitoring these cellular and molecular factors in the healing wounds of our different knockout mice, we highlighted a complex picture that is schematically summarized in Figure 8. The results detected a group of factors, including neutrophils, IL-6, α-SMA, VEGF, and TNFα, which appeared to be homogenously decreased in the healing tissue of all single-KO mice. These data suggest that these factors play a key role in the contribution given by the ICOS/ICOSL/OPN network to wound healing.

We did not detect a decrease in vessel formation in the single-KO strains, and we observed a decrease in T cells and macrophages only in ICOSL-KO mice, which is discordant with the Maeda data. Nevertheless, our data pointed out the crucial role played by blood vessels by showing that they were strikingly increased in both double-KO strains, where they may be involved in normalizing wound closure. In ICOS/OPN-KO mice, this vessel increase might be supported by VEGF and the transient elevated levels of TNFα [21] whereas, in ICOSL/OPN-KO mice, it may be supported by factors produced by the abundant infiltration of neutrophils, macrophages, and T cells.

An intriguing point is that the healing tissue of the two double-KO strains displayed partly different cellular and molecular features, including collagen deposition, neutrophil and macrophage infiltration, VEGF and TNFα expression, which suggests that the scheme shown in Figure 7 might be complicated by other interactors of ICOS or ICOSL that are not included in the figure. For instance, a role may be played by the αvβ3 integrin that seems to be capable of interacting not only with OPN but also with ICOSL [20].

A similar compensatory effect has been previously reported for IL-6 and its receptor IL-6Rα since mice deficient in IL-6 display defective wound healing that is surprisingly rescued in mice that are deficient in both IL-6 and IL-6Rα showing increased macrophage infiltration and angiogenesis. Moreover, in this system, multiple molecular interactions may be involved since IL-6Rα interacts with the signaling transmembrane protein gp130 that is shared with several other surface receptors [22].

A second point is that the treatment with ICOS-Fc improves healing in all mouse strains expressing ICOSL (i.e., WT, ICOS-KO, ICOS/OPN-KO) but it has no effect in those lacking ICOSL (i.e., ICOSL-KO and ICOSL/OPN-KO), indicating that the effect depends on the ICOS-Fc-mediated triggering of ICOSL and not on the inhibition of the endogenous ICOS or the triggering of Fc-gamma receptors (FcγRs).

Monitoring the effect of ICOS-Fc on cellular and molecular factors in the healing tissue of the different mouse strains detected a complex picture as schematically depicted in Figure 9.

The first finding was that ICOS-Fc does not induce any effect on these factors in the mouse strains lacking ICOSL, which is in line with the data on wound closure. A second finding was the detection of a group of factors, including vessels, macrophages, and the TREM2/TREM1 expression ratio, that were increased following the ICOS-Fc treatment in all mouse strains expressing ICOSL, indicates that these effects are directly mediated by the triggering of ICOSL without substantial influences exerted by the endogenous OPN and ICOS. In contrast, a variegate picture was detected by looking at the other factors, suggesting that these effects involve variable contributions from the endogenous OPN and/or ICOS.

## 4. Materials and Methods

### 4.1. Mice

C57BL/6J, Knockout (KO) B6.129P2-Icos^tm1Mak^/J (ICOS-KO), B6.129P2-Icosl^tm1Mak^/J (ICOSL-KO), and B6.129S6 (Cg)-Spp1^tm1Blh^/J (OPN-KO) mice (The Jackson Laboratory, Bar Harbor, ME, USA) were bred under pathogen-free conditions in the animal facility of the Università del Piemonte Orientale, Department of Health Sciences (Authorization n° 217/2020-PR), and all procedures were conducted in accordance with the Ethical Committee and European guidelines. The double-KO mice (ICOSL/OPN-KO and ICOS/OPN-KO) were generated by backcrossing the single-KO strains (Authorization n° 831/2020-PR).

### 4.2. Wound Healing

The day before wound induction, the mice were anesthetized with 2% isoflurane and their backs were shaved to prevent any interference between the shaving procedure and the wounds. The day after, the mice were anesthetized and the wounds were induced on their backs using a 4-mm puncher (Kai Medical, Solingen, Germany). The wound areas were photographed and measured using the following formula: (a/2) × (b/2) × 3.14, where a and b are the two perpendicular diameters. The wound closure was calculated using the following formula: (Wound Area^T0^ − Wound Area^TX^)/Wound Area^T0^ × 100. The mice were treated daily with 10 µg ICOS-Fc in PBS (Sigma-Aldrich, St. Louis, MO, USA) directly on the wound site and the control groups were treated with an equal volume of PBS. The mice were monitored daily until the complete wound closure. In some experiments, the mice were sacrificed at days 1, 2, 3, and 4 post-wound induction for RT-PCR and histological analysis.

### 4.3. Real-Time PCR Analyses

Total RNA was isolated from skin samples collected at 1, 2, and 3 days (T1, T2, and T3, respectively) after the injury, using the TRIzol reagent (Sigma-Aldrich, St. Louis, MO, USA). RNA (1 µg) was retrotranscribed using the QuantiTect Reverse Transcription Kit (Qiagen, Hilden, Germany). The expression levels of IL-6, TNFα, α-SMA, TREM1, TREM2, and VEGF were evaluated using a gene expression assay (Assay-on-Demand; Applied Biosystems, Foster City, CA, USA). The β-actin gene was used to normalize the cDNA amounts. Real-time PCR was performed using the CFX96 System (Bio-Rad Laboratories, Hercules, CA, USA) in duplicate for each sample in a 10 µL final volume containing 1 µL of diluted cDNA, 5 µL of the TaqMan Universal PCR Master Mix (Applied Biosystems, Foster City, CA, USA), and 0.5 µL of the Assay-on-Demand mix. The results were analyzed with a ΔΔ threshold cycle method.

### 4.4. Histological Analyses

The skin samples were collected at 3 and 4 days (T3 and T4) after the injury and were processed for paraffin embedding. The samples were cut to a thickness of 4 µm and used for the histological analyses. Samples were stained with hematoxylin and eosin (Sigma-Aldrich, St. Louis, MO, USA) for tissue morphology and fibroblasts area evaluation or with picrosirius red (Abcam, Cambridge, UK) to evaluate the fibrotic tissue production, following the manufacturer’s instructions. The immunohistochemical (IHC) analysis was used to evaluate CD31, MPO, CD3, and F4/80 expression in the wound site to analyze new vessel formation and the infiltration of immune cells (neutrophils, T cells, and macrophages). Samples were treated with a citrate buffer (Vector Laboratories, Susteren, Netherlands) for antigen retrieval and endogenous peroxidases were blocked with 3% H_2_O_2_ (Sigma-Aldrich, St. Louis, MO, USA). To avoid the non-specific binding of the secondary antibody, the samples were incubated with 5% normal goat serum (NGS) (Sigma-Aldrich, St. Louis, MO, USA) for 1 h at room temperature (RT). Samples were incubated with CD31 (Abcam, Cambridge, UK, 1:50), MPO (Invitrogen, Waltham, MA, USA, 1:100), CD3 (Invitrogen, Waltham, MA, USA, 1:150) or F4/80 (Invitrogen, Waltham, MA, USA, 1:100) primary antibody overnight at 4 °C. After washing, the samples were treated with a goat anti-rabbit horseradish peroxidase (HRP)-conjugated secondary antibody (Sigma-Aldrich, St. Louis, MO, USA). The samples were then treated with 3,3′-diaminobenzidine (DAB) (Agilient Technologies, Santa Clara, CA, USA) to visualize the reaction with HRP. Following IHC, the samples were subsequently counterstained with hematoxylin (Sigma-Aldrich, St. Louis, MO, USA), dehydrated, and mounted with cover slips. The slides were acquired by Pannoramic MIDI (3D Histech, Budapest, Hungary) and the images were taken for each slide at 20× or 40× magnification. The positive area used to calculate the CD31 expression, fibroblast migration, and collagen production was calculated using the following formula: (Positive Area/Total Area) × 100%. The MPO, CD3, and F4/80 positive cells were counted in the wound site and expressed as the number/field of 15 20× fields for each sample.

### 4.5. Statistical Analyses

Statistical analyses were performed using the Mann–Whitney test or the student t-test using the GraphPad Instat Software 3.06 (GraphPad Software, San Diego, CA, USA). Data are expressed as the mean and standard error of the mean (SEM) and statistical significance was set at *p* < 0.05.

## 5. Conclusions

In conclusion, the involvement of the ICOS/ICOSL/OPN network in skin wound healing requires a balanced activity of the three molecules, as wound closure is delayed in single-KO mice but occurs normally in double-KO mice. In all single-KO mice, the defect appears to involve the impaired recruitment of neutrophils and the defective expression of α-SMA, IL-6, TNFα, and VEGF. However, the normalization of wound closure in double-KO mice does not simply depend on the normalization of these factors, but rather on an increase in vessel formation. This increase in angiogenesis, possibly supported by M2 macrophages, seems to be the main mechanism by which the treatment with ICOS-Fc promotes wound healing through the triggering of ICOSL. Noteworthy, this proangiogenic effect contrasts with that observed in tumors, where ICOS-Fc is primarily anti-angiogenic, suggesting that angiogenesis is driven by partly different mechanisms in wound healing compared to cancer development [15]. For instance, it might be relevant that ICOS-Fc has no effect on the in vitro angiogenesis induced by VEGF, whereas it inhibits that induced by OPN [9,15]. The positive effects of ICOS-Fc in tissue repair are also supported by the previous findings that the treatment with ICOS-Fc favors liver repair after the acute damage induced by CCL_4_ [18], counteracts organ injury in sepsis [23], and protects from the development of osteoporosis [12], which opens the way to exploit ICOS-Fc as a powerful immunoregulatory tool to improve tissue repair in different contexts of tissue injury. Noteworthy, an important area for future research is to understand the differences in angiogenesis mechanisms between wound healing and tumor progression. A deeper understanding of these processes could lead to the development of ICOS-Fc derivatives or combinations that promote wound healing without promoting tumor growth, providing safer, targeted therapies for cancer patients in need of tissue repair. However, a limitation of all these studies is that they are focused on the C57BL/6 strain, that is the standard strain used for KO mice, and experiments should be extended to other mouse strains and, possibly, humans.

## Figures and Tables

**Figure 1 ijms-25-12390-f001:**
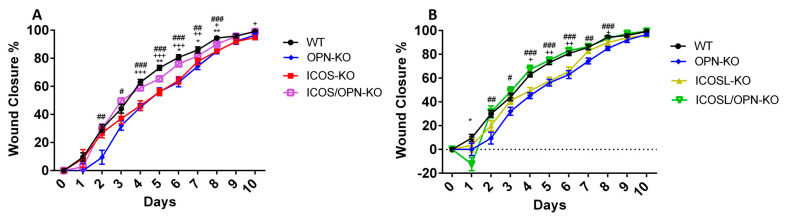
The wound closure rate in mice with different genotypes: osteopontin knock out (OPN-KO), Inducible T-cell costimulator knock out (ICOS-KO), ICOS ligand knock out (ICOSL-KO), ICOS/OPN-KO and ICOSL/OPN-KO. (**A**) The graph shows the wound closure rate of WT (n = 21) compared to OPN-KO (n = 19), ICOS-KO (n = 16), and ICOS/OPN-KO (n = 24). WT vs. ICOS/OPN-KO: * *p* < 0.05; ** *p* < 0.005; WT vs. ICOS-KO: + *p* < 0.05; ++ *p* < 0.01; +++ *p* < 0.0001; WT vs. OPN-KO: # *p* < 0.05; ## *p* < 0.005; ### *p* < 0.001. (**B**) The graph shows the wound closure rate of WT (n = 21) compared to OPN-KO (n = 19), ICOSL-KO (n = 18), and ICOSL/OPN-KO (n = 17). WT vs. ICOSL/OPN-KO: * *p* < 0.005; WT vs. ICOSL-KO: + *p* < 0.005; ++ *p* < 0.001; WT vs. OPN-KO: # *p* < 0.05; ## *p* < 0.005; ### *p* < 0.001. The graphs show the % of wound closure calculated using the following formula: (Wound AreaT0-Wound AreaTX)/Wound AreaT0 × 100. The statistical analysis is calculated by the Mann–Whitney test.

**Figure 2 ijms-25-12390-f002:**
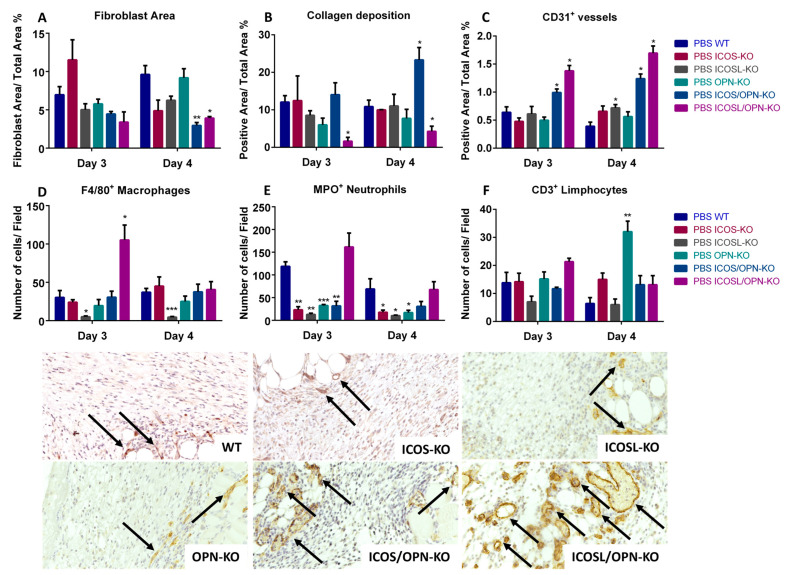
A comparison between WT mice and KO mice in terms of fibroblast infiltration, collagen deposition, new vessel formation (CD31^+^), macrophages (F4/80^+^), neutrophils (MPO^+^), and T cell (CD3^+^) infiltration in the wound bed (IHC). Representative images of CD31 day 3 staining are reported (20×), the arrows indicate the positive signal in correspondence of the blood vessels, n = 5 for each group. * *p* < 0.05; ** *p* < 0.005; *** *p* < 0.001 are calculated by the Mann–Whitney test.

**Figure 3 ijms-25-12390-f003:**
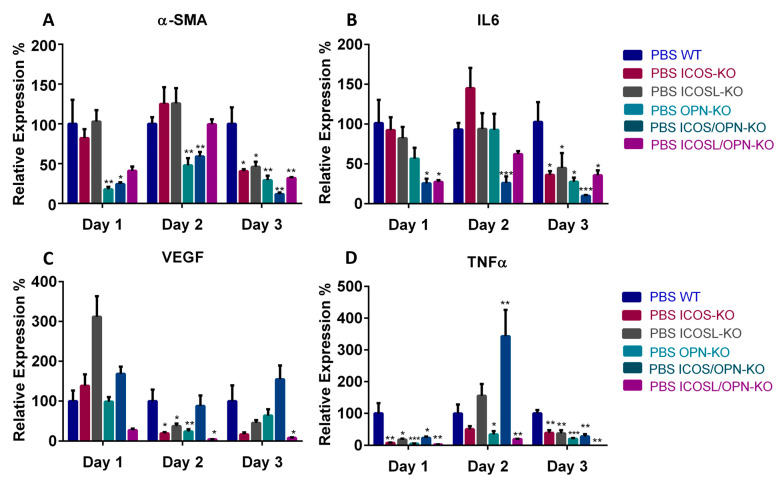
A comparison between KO mice and WT mice in terms of the expression of relevant markers during the wound healing in mice treated with PBS. The graphs show the relative expression calculated by RT-PCR of different relevant markers in the wound healing: IL-6, TNFα, CD31, α-SMA, and TREM2/TREM1 ratio. (PBS n = 18; ICOS-Fc n = 18) vs. PBS WT: * *p* < 0.05; ** *p* < 0.005; *** *p* < 0.001 are calculated by the Mann–Whitney test.

**Figure 4 ijms-25-12390-f004:**
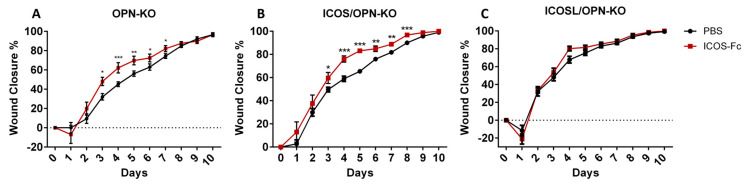
The effect of the ICOS-Fc treatment on wound closure in OPN-KO, ICOS/OPN-KO, and ICOSL/OPN-KO mice. The graphs show the % of wound closure in (**A**) OPN-KO mice (n = 28–11), (**B**) ICOS/OPN-KO mice (n = 24–9), and (**C**) ICOSL/OPN-KO mice (n = 14–10) following the treatment with PBS or ICOS-Fc. The graphs show the % of wound closure calculated as previously described in Figure 2. * *p* < 0.05; ** *p* < 0.005; *** *p* < 0.001 are calculated by the Mann–Whitney test.

**Figure 5 ijms-25-12390-f005:**
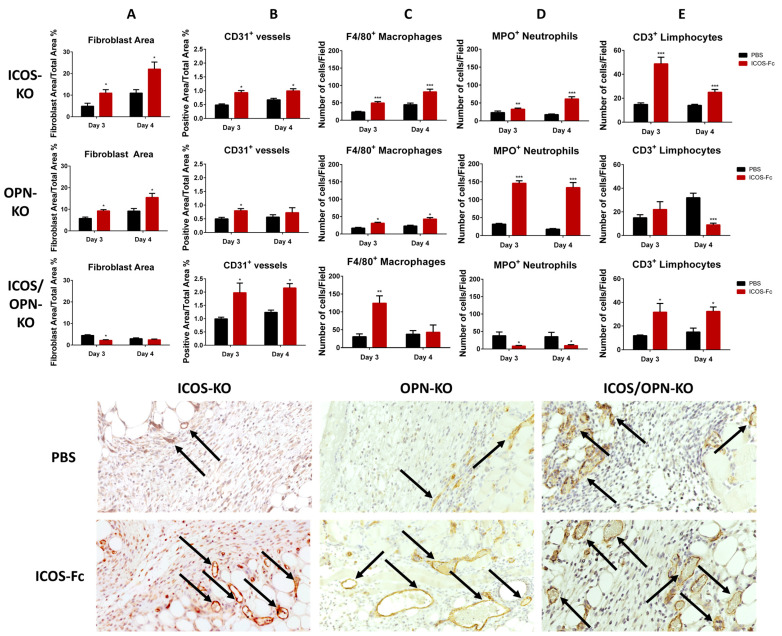
A comparison between WT mice and KO mice treated with PBS or ICOS-Fc in terms of fibroblast infiltration, collagen deposition, new vessel formation (CD31), macrophages (F4/80), neutrophils (MPO), and T cell (CD3) infiltration in the wound bed (IHC). n = 5 for each group * *p* < 0.05; ** *p* < 0.005; *** *p* < 0.001 are calculated by the Mann–Whitney test for the fibroblast area and CD31^+^ vessels and unpaired *t*-test for macrophages (F4/80), neutrophils (MPO), and T cell (CD3) infiltration. Representative images of CD31 day 3 staining are reported (20×), the arrows indicate the positive signal in correspondence of the blood vessels.

**Figure 6 ijms-25-12390-f006:**
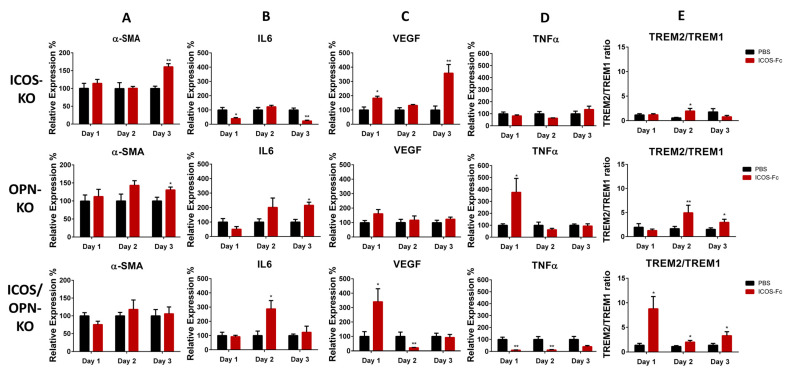
A comparison between KO mice in terms of the expression of relevant markers during the wound healing in mice treated with PBS and ICOS-Fc. The graphs show the relative expression calculated by RT-PCR of different relevant markers in the wound healing: IL-6, TNFα, CD31, α-SMA, and TREM/1 ratio. (PBS n = 18; ICOS-Fc n = 18). * *p* < 0.05; ** *p* < 0.005 are calculated by the Mann–Whitney test, except for the TREM2/TREM1 ratio that was calculated by the unpaired test.

**Figure 7 ijms-25-12390-f007:**
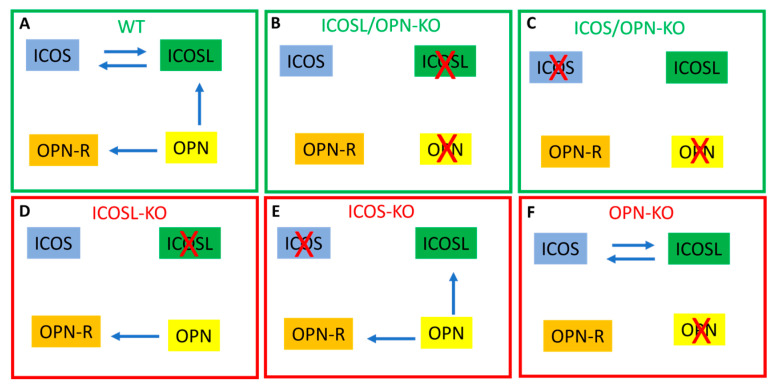
Scheme of the signaling pathways triggered by the ICOS/ICOSL/OPN network in the mouse strains showing normal (green) or defective (red) wound healing. The blue arrows indicate the active signaling pathways; the red crosses indicate the defective molecules of the network; OPN-R indicate the OPN receptors other than ICOSL and include several integrins and CD44.

**Figure 8 ijms-25-12390-f008:**
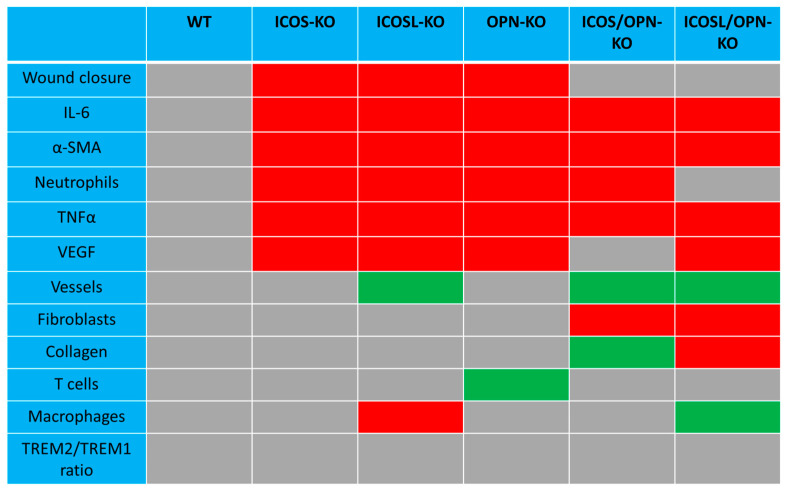
Scheme of the differences between the wild type mice and each knockout strain in the wound healing closure and cellular and molecular healing markers. Grey boxes: not different compared to wild type mice (WT); red boxes: decreased compared to WT mice; green boxes: increased compared to WT mice.

**Figure 9 ijms-25-12390-f009:**
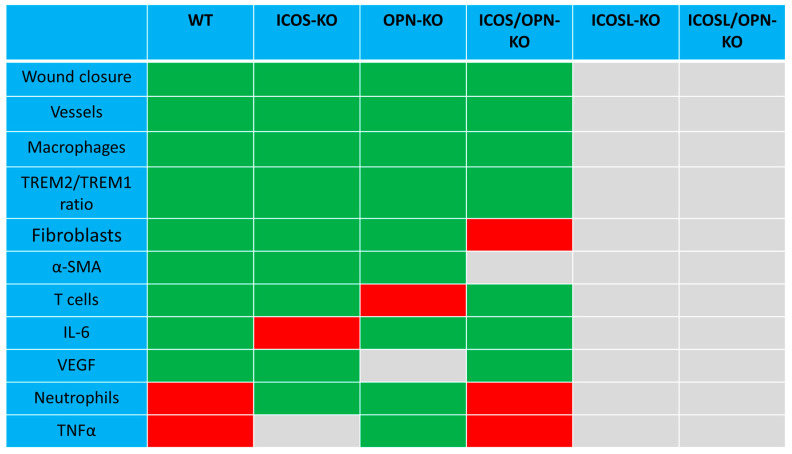
Scheme of the effects exerted by ICOS-Fc on the wound healing closure and cellular and molecular healing markers in wild type and knockout mice. Grey boxes: no effect induced by ICOS-Fc; red boxes: decrease induced by ICOS-Fc; green boxes: increase induced by ICOS-Fc.

## Data Availability

Raw data are available on request from the corresponding author.

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
