# Peer review of "Role of Balanced Involvement of the ICOS/ICOSL/Osteopontin Network in Cutaneous Wound Healing"

_ijms, 2024, doi:10.3390/ijms252212390_

Round 1
Reviewer 1 Report
Comments and Suggestions for Authors
1. What might be the underlying molecular mechanism that leads to normalization of wound healing in double-knockout mice? Could other molecules compensate for the loss of both ICOS and OPN?
2. The discussion mentions contrasting effects of ICOS-Fc on angiogenesis in wound healing and tumors. Is it possible to speculate on the specific factors that might account for these differences?
3. Were there any limitations in this study such as sample size or strain differences, that might affect the generalizability or reproducibility of the results ?
4. The study compares single and double-knockout models, but the mechanisms underlying the compensatory effects in double-knockout strains could be explored further. However, it is suggested to add references to similar compensatory phenomena in literature in order to strengthen the discussion.
Author Response
Comments 1: What might be the underlying molecular mechanism that leads to normalization of wound healing in double-knockout mice? Could other molecules compensate for the loss of both ICOS and OPN?
Response 1: The key molecular mechanism that clearly seems to compensate the wound healing process in both double KO mice is the increase of vessel formation, as stated in Abstract (lines 27-28), Introduction (lines 87-88), Discussion (lines 281-283), and Conclusions (lines 386-388). Moreover, in Discussion (lines 283-286), we suggest that “In ICOS/OPN-KO mice, this vessel increase might be supported by VEGF and the transient elevated levels of TNFα whereas, in ICOSL/OPN-KO mice, it may be supported by factors produced by the abundant infiltration of neutrophils, macrophages and T cells”.
Comments 2: The discussion mentions contrasting effects of ICOS-Fc on angiogenesis in wound healing and tumors. Is it possible to speculate on the specific factors that might account for these differences?
Response 2: In the Conclusions, where we state that “this proangiogenic effect contrasts with that observed in tumors, where ICOS-Fc is primarily anti-angiogenic, suggesting that angiogenesis is driven by partly different mechanisms in wound healing compared to cancer development”, we added the sentence “For instance, it may be relevant that ICOS-Fc has no effect on angiogenesis induced in vitro by VEGF whereas it inhibits that induced by OPN [9,15].” (lines 393-395)
Comments 3: Were there any limitations in this study such as sample size or strain differences, that might affect the generalizability or reproducibility of the results?
Response 3: We consider the sample size of our experimental group adequate and in accordance with the experimental protocol approved by the Italian Ministry of Health for protocol (No. DB064.80). The C57BL/6J is the necessary mouse strain as control for the KO strains that we used for the experiments (The Jackson Laboratory, Bar Harbor, ME, USA). To underline this limitation, we added the sentence “However, a limitation of all these studies is that they are focused on the C57BL/6 strain, that is the standard strain use for KO mice, and experiments should be extended to other mouse strains and, possibly, humans” In the Conclusions (line 404-406).
Comments 4: The study compares single and double-knockout models, but the mechanisms underlying the compensatory effects in double-knockout strains could be explored further. However, it is suggested to add references to similar compensatory phenomena in literature in order to strengthen the discussion.
Response 4: In the Discussion, we added the following sentence “A similar compensatory effect has been previously reported for IL-6 and its receptor IL-6Rα, since mice deficient for IL-6 display defective wound healing that is surprisingly rescued in mice dual deficient for IL-6 and IL-6Rα showing increased macrophage infiltration and angiogenesis. In this system, too, multiple molecular interactions may be involved since IL-6Rα interacts with the signaling transmembrane protein gp130 that is shared with several other surface receptors”. [Molly M. McFarland-Mancini, Holly M. Funk, Andrew M. Paluch, Mingfu Zhou, Premkumar Vummidi Giridhar, Carol A. Mercer, Sara C. Kozma, Angela F. Drew; Differences in Wound Healing in Mice with Deficiency of IL-6 versus IL-6 Receptor. J Immunol 15 June 2010; 184 (12): 7219–7228].
Reviewer 2 Report
Comments and Suggestions for Authors
The current study investigated the involvement of the Inducible T cell Co-stimulator/Inducible T cell Co-stimulator ligand/osteopontin (ICOS/ICOSL/OPN) network in skin wound healing by analyzing mice that are single knockouts (KO) for ICOS, ICOSL, or OPN, or double knockouts for ICOS/OPN or ICOSL/OPN.
The study demonstrated that all components of the ICOS/ICOSL/OPN network are involved in skin wound healing and the deficiency of each of them impairs healing process as evidenced in ICOS-KO, ICOSL-KO, and OPN-KO mice. Additionally, The study showed that the wound healing closure was normal in ICOS/OPN-KO and ICOSL/OPN-KO mice and indicated that the combination of the OPN deficiency with the deficiency of either ICOS or ICOSL has compensatory effects on skin wound healing. The study confirmed that the involvement of the ICOS/ICOSL/OPN network in skin wound healing requires a balanced activity of the three molecules, as wound closure is delayed in single-KO mice but occurs normally in double-KO mice.
The manuscript is clearly presented, well written and the results are worthy of publication, however there are some issues that should be addressed prior to acceptance
Major comments
1- Representative histological images related to the results shown in Fig. 2 should be added.
2- Why the authors only selected day 3 and day 4 in the comparison between WT mice and KO mice in terms of fibroblast infiltration, collagen deposition, new vessel formation (CD31+ ), macrophages (F4/80+ ), neutrophils (MPO+ ) and T cell (CD3+ ) infiltration in the wound bed
3- What is the possibility of adding Westren blot analyses for IL6, TNFα, CD31, VEGF, α-SMA in Fig. 3 and Fig. 6 ???
4- Representative histological images related to the results shown in Fig. 5 should be added.
Minor comments
1- The ICOS, ICOSL abbreviations should be defined in the manuscript
Author Response
Comments 1: Representative histological images related to the results shown in Fig. 2 should be added.
Response 1: Representative histological images of day 3 CD31 staining have been added to figure 2.
Comments 2: Why the authors only selected day 3 and day 4 in the comparison between WT mice and KO mice in terms of fibroblast infiltration, collagen deposition, new vessel formation (CD31+ ), macrophages (F4/80+ ), neutrophils (MPO+ ) and T cell (CD3+ ) infiltration in the wound bed
Response 2: We decided to focus on early stages of wound healing to allow the comparisons with our previous work (Stoppa et al., 2022) and the work by Maeda et al., showing striking differences between WT and ICOS-KO or ICOSL-KO mice.
Comments 3: What is the possibility of adding Western blot analyses for IL6, TNFα, CD31, VEGF, α-SMA in Fig. 3 and Fig. 6 ???
Response 3: Unfortunately, all the biological material obtained from the experiments has been used for RNA extraction to perform RT-PCR and for histological analysis. However, our previous experience on tumors showed that differences are hardly detected by western blot.
Comments 4: Representative histological images related to the results shown in Fig. 5 should be added.
Response 4: Representative histological images of PBS and ICOS-Fc groups at day 3 of CD31 staining have been added to figure 5.
Comments 5: The ICOS, ICOSL abbreviations should be defined in the manuscript
Response 5: ICOS and ICOSL have been defined
Round 2
Reviewer 2 Report
Comments and Suggestions for Authors
The authors have properly addressed the reviewer comments from the previous review round and they have made satisfactory revisions to the manuscript. Therefore, reviewer recommends acceptance of the revised manuscript for publication